# Statins Do Not Directly Inhibit the Activity of Major Epigenetic Modifying Enzymes

**DOI:** 10.3390/cancers11040516

**Published:** 2019-04-10

**Authors:** Stephanie Bridgeman, Wendy Northrop, Gaewyn Ellison, Thiru Sabapathy, Phillip E. Melton, Philip Newsholme, Cyril D. S. Mamotte

**Affiliations:** 1School of Pharmacy and Biomedical Sciences, and Curtin Health Innovation Research Institute, Curtin University, Bentley, WA 6102, Australia; stephanie.allen@postgrad.curtin.edu.au (S.B.); wendy.northrop@student.curtin.edu.au (W.N.); gaewyn.ellison@postgrad.curtin.edu.au (G.E.); t.sabapathy@postgrad.curtin.edu.au (T.S.); phillip.melton@curtin.edu.au (P.E.M.); Philip.Newsholme@curtin.edu.au (P.N.); 2Centre for Genetic Origins of Health and Disease, Faculty of Health and Medical Science, the University of Western Australia, Perth, WA 6000, Australia

**Keywords:** HMG-CoA reductase inhibitors, epigenetics, histone acetylation, DNA methylation, HDAC inhibitors, DNMT inhibitors

## Abstract

The potential anticancer effects of statins—a widely used class of cholesterol lowering drugs—has generated significant interest, as has the use of epigenetic modifying drugs such as HDAC and DNMT inhibitors. We set out to investigate the effect of statin drugs on epigenetic modifications in multiple cell lines, including hepatocellular carcinoma, breast carcinoma, leukemic macrophages, cervical adenocarcinoma, and insulin-secreting cells, as well as liver extracts from statin-treated C57B1/6J mice. Cells or cell extracts were treated with statins and with established epigenetic modulators, and HDAC, HAT, and DNMT activities were quantified. We also examined histone acetylation by immunoblotting. Statins altered neither HDAC nor HAT activity. Accordingly, acetylation of histones H3 and H4 was unchanged with statin treatment. However, statins tended to increase DNMT activity. These results indicate that direct inhibition of the major classes of epigenetic modifying enzymes, as previously reported elsewhere, is unlikely to contribute to any anticancer effects of statins. This study concerned global effects on epigenetic enzyme activities and histone acetylation; whether statins influence epigenetic modifications in certain genomic regions, cannot be ruled out and remains to be investigated.

## 1. Introduction

HMG-CoA reductase inhibitors, commonly known as statins, are widely used to reduce low-density lipoprotein (LDL) cholesterol, with an estimated >30 million statin users in the United States alone [1]. By inhibiting HMG-CoA reductase, the key rate-determining step in the mevalonate pathway of cholesterol biosynthesis, statins increase the uptake of LDL from the bloodstream into cells, primarily hepatocytes, thus lowering plasma LDL. Multiple large-scale clinical trials have established the efficacy of statins in reducing morbidity and mortality from cardiovascular disease (CVD). A meta-analysis of 59 placebo-controlled trials encompassing 157,217 participants found statins reduced major coronary events by 31% and overall mortality by 13% [2]. In addition to lipid-lowering effects, statins have also been reported to have beneficial pleiotropic effects, including anti-inflammatory, anti-proliferative, and immunomodulatory activities, which may protect against cancer [3]. Meta-analyses of cohort and case–control studies found statin use is associated with a reduced risk of multiple types of cancers, including liver, colorectal, gastric, ovarian, hematological, and esophageal cancers [4], although no such association was found in a meta-analysis restricted to randomized controlled trials [5]. Furthermore, a meta-analysis of 95 cohort studies found that amongst cancer patients, statin use decreased all-cause and cancer-specific mortality, including amongst prostate, colorectal, and breast cancer patients [6]. In a recent review, we described the evidence that statins have epigenetic effects which may contribute to anticancer activity [7].

In recent years, there has been a plethora of research on epigenetic modifications and their association with cancer and other common disorders such as CVD and diabetes [8,9,10]. Epigenetic modifications are changes to DNA that may alter gene expression without altering the base nucleotide code. The two major classes of epigenetic modifications are histone modifications, of which the most studied and best understood is histone acetylation, and DNA methylation. These modifications are carried out by epigenetic modifying enzymes. Histone acetylation, generally associated with increased gene expression, is controlled by two opposing classes of enzymes: histone acetyltransferases (HATs) and histone deacetylases (HDACs). DNA methyltransferases (DNMTs) transfer methyl groups from S-adenosyl methionine to nucleotides, primarily cytosines in CpG dinucleotides, in a process usually associated with transcriptional silencing. There are several reports of statins inhibiting HDAC enzyme activity [11,12,13] and a single report of DNMT inhibition [14], with the majority of these studies linking such effects to anticancer activity. However, notable gaps remain in the literature. For example, previous reports of an inhibitory effect on HDACs were based on direct enzymatic inhibition of recombinant HDACs or on nuclear extracts, and findings on histone acetylation have been mixed [12,13]. Importantly, the effect of statins on HDAC activity in live cells has not been reported, despite being more relevant to in vivo and treatment models, nor has the effect of statins on HAT activity.

We therefore conducted a comprehensive investigation of the effects of statins on HDAC activity in both live cells and nuclear extracts, using a variety of different cell lines, including HepG2 hepatocellular carcinoma cells, MDA-MB-231 breast carcinoma cells, and HeLa cervical adenocarcinoma extracts. Since statin use has also been associated with diabetes [15], we also used BRIN-BD11 cells, a model of insulin-secreting rat pancreatic β-cells, and the livers of statin-treated mice, both on a control diet and on a high-fat diet. By using a variety of different cell types, we consider our findings are generally applicable, and there is little evidence to suggest that enzymatic properties of HDACs differ between different cell types. By applying statins directly to cell extracts we could determine if statins inhibit activity through direct interaction with the enzyme, as previously reported for HDAC activity by Lin et al. [12]. On the other hand, by measuring enzymatic activity in treated cells, as previously reported with DNMT activity by Kodach et al. [14], we could determine if statins alter activity either directly or through altered expression of enzymes or other indirect phenotypic changes. For example, the established HDAC inhibitor trichostatin A (TSA) directly inhibits HDAC enzymes through interaction with the active site of HDAC enzymes [16], while the antidiabetic drug metformin is thought to reduce HDAC activity not through direct inhibition, but indirectly through activation of AMP kinase, which subsequently phosphorylates HDAC proteins and thus reduces their enzymatic activity [17]. We also investigated effects on global histone acetylation in treated cells and HAT and DNMT activity in cell extracts.

## 2. Results

### 2.1. The Effect of Statins on Cell Viability and Cholesterol

Dose-response curves for viability and measurement of cellular cholesterol reduction were first conducted to ensure use of appropriate statin doses for later epigenetic studies in live cells. Statins decreased HepG2 cell viability, as assessed by the alamarBlue^®^ assay, in a dose-dependent manner (Figure 1a). This was most apparent for the lipophilic atorvastatin and simvastatin, reducing viability to 13% (±5% SEM) and 3% (±5% SEM), respectively, at 500 µM after 24 h. The hydrophilic pravastatin and rosuvastatin were less toxic, with viability remaining at 70% (±6% SEM) and 83% (±17% SEM), respectively, at 500 µM. Simvastatin reduced viability at all tested concentrations, however no other statin affected viability at 10 µM in HepG2 cells, and no statin affected viability in MDA-MB-231 breast carcinoma cells at 10 µM (Appendix A). Furthermore, all statins significantly lowered HepG2 cellular cholesterol at 10 µM after 24 h treatment (Figure 1b), ranging from 33% (±14% SEM) lower cholesterol by pravastatin to 53% (±8 SEM) by simvastatin, demonstrating that 10 µM is a sufficient concentration for HMG-CoA reductase inhibition by statins. As such this concentration was used for subsequent experiments.

### 2.2. The Effect of Statins on HDAC Activity in Cultured Cells

No statin (at 10 µM) inhibited HDAC activity in live HepG2 cells after 24 h treatment (Figure 2a). Conversely, sodium butyrate and trichostatin A (TSA), both established HDAC inhibitors, significantly inhibited HDAC activity (*p* < 0.0001). Atorvastatin, chosen as a model statin due to its potency and widespread use, was tested at a range of concentrations and did not inhibit HDAC activity at up to and including 50 µM (Appendix A), a concentration sufficiently high to impact cell viability, and more than sufficient to reduce cellular cholesterol content. Statins similarly failed to inhibit HDAC activity in MDA-MB-231 breast carcinoma cells and BRIN-BD11 insulin-secreting cells, while sodium butyrate and TSA showed significant inhibition (*p* < 0.0001) (Figure 2b,c). 

Since statins were reported to directly inhibit HDAC activity in A549 lung carcinoma nuclear extracts [12], we examined the direct effect of statins on HDAC activity in HepG2 nuclear extracts. As in live cells, atorvastatin and pravastatin did not inhibit HDAC activity at 100 µM (Figure 2d), a dose significantly higher than the 10 µM used by Lin et al. [12]. By contrast, and as in live cells, TSA strongly inhibited HDAC activity (*p* < 0.0001). A similar lack of effect was found with statins in MDA-MB-231 nuclear extracts (Appendix A). 

Additionally, experiments were conducted using the Fluor De Lys^®^ HDAC fluorometric activity assay kit, also used in previous studies by Lin et al. [12], Chen et al. [13], and Singh et al. [11]. HDAC inhibition was observed with TSA (*p* < 0.0001) and sodium butyrate (*p* < 0.0001) but not with statins in either HepG2 nuclear extracts or in the HeLa cervical adenocarcinoma nuclear extract provided with the kit (Figure 3).

### 2.3. HDAC Activity in the Liver of Statin-Treated Mice

We next utilized liver samples from a previous animal study by our group in which male C57B1/6J mice were fed normal chow or a high-fat diet and treated with atorvastatin for 12 weeks. HDAC activity was measured in nuclear extracts isolated from the liver of control animals fed a normal or high-fat diet and from atorvastatin-treated animals fed a normal or high-fat diet. As for studies on live cells and nuclear extracts, we found no effect with statin treatment on HDAC activity compared to the control (Figure 4). Furthermore, there was no effect of diet on HDAC activity.

### 2.4. The Effect of Atorvastatin on HAT Activity in HepG2 Nuclear Extracts

In addition to HDAC inhibition, compounds may alter histone acetylation levels through influences on HATs. However, a 100 µM dose of atorvastatin had no effect on HAT activity when applied directly to HepG2 nuclear extracts (Figure 5). Conversely, and as expected, the control HAT inhibitor curcumin reduced the HAT activity of HepG2 nuclear extracts by 70% (±6% SEM).

### 2.5. The Effects of Atorvastatin Treatment on Global Histone Modifications

Baseline levels of histone H3K9 and H4 acetylation were low in HepG2 cells (Figure 6a,b). As expected, acetylation was increased in a dose-dependent manner by sodium butyrate treatment. Conversely, acetylation levels remained at a low baseline level in cells treated with atorvastatin. Similar results were seen in BRIN-BD11 cells (Appendix A). Baseline levels of histone acetylation were higher in THP-1 leukemic macrophages and were further increased by 5 mM sodium butyrate treatment, but atorvastatin had no effect (Figure 6c,d). 

In order to ascertain if the lack of lipoproteins in the media affected results, experiments were also conducted in HepG2 cells treated in high-lipoprotein serum (HLPS). The results obtained in HLPS were almost identical to those in lipoprotein-deficient serum (LPDS), with sodium butyrate drastically increasing H3K9 acetylation, and atorvastatin having no effect (Appendix A).

### 2.6. The Effect of Atorvastatin on DNMT Activity

Statins have also been reported to inhibit DNMT activity [14]. When applied directly to HepG2 whole cell extracts, 200 µM atorvastatin did not significantly inhibit, and in fact tended to increase, DNMT activity (Figure 7a). Conversely, curcumin, a known DNMT inhibitor, decreased DNMT activity by 65% (±8% SEM). This was significantly greater than equal dose of the rationally designed DNMT inhibitor RG108, which inhibited DNMT activity by 25% (±9% SEM) and was just above the threshold for significance (*p* = 0.057). Similar results were obtained with BRIN-BD11 cell extracts (Appendix A). 

DNMT activity was also measured in the extracts of cells that had been treated for 24 h with atorvastatin or curcumin. DNMT activity was quite variable in these experiments but DNMT activity of cells treated with atorvastatin was in fact higher than the DMSO vehicle control (*p* < 0.01) (Figure 7b). In extracts of curcumin treated cells, DNMT activity was not significantly different than that of the control, possibly due to the rapid degradation of curcumin in media [18].

## 3. Discussion

In view of their widespread use and tolerability, the potential anticancer effects of statins held considerable appeal. In vitro studies on cancer cell lines have revealed potential mechanisms for anticancer activity, including cell cycle arrest, induction of apoptosis and prevention of metastasis [19]. It had also been suggested that inhibition of epigenetic modifying enzymes, such as HDACs and DNMTs, may contribute to the anticancer activity of statins, as we reviewed previously [7]. Epigenetic modifications are often altered in cancer cells and upregulation of HDACs and DNMTs have been associated with numerous cancers [10]. As a result, multiple epigenetic modifying drugs have been approved for cancer treatment, including the HDAC inhibitors vorinostat, romidepsin, belinostat, panobinostat, and chidamide, and the DNMT inhibitors azacytidine and decitabine [20]. However, these are associated with severe side effects and treatment-related deaths have been reported in clinical trials [21]. Thus, based on previous studies, statins offered potential benefits over current epigenetic drugs for cancer. However, our results indicate that direct inhibition of major epigenetic enzymes is unlikely to contribute to any anticancer effects of statins.

Studies of various cancer cell lines [12,22], a macrophage line [23], and a rabbit model of atherosclerosis [24] reported that statin treatment can cause hyperacetylation of histones H3 and H4, either globally or in certain genomic regions, or inhibition of HDAC activity and expression. Our extensive results on HDAC activity in live cells are in complete contrast to these previous studies. This was demonstrated on a variety of different cells lines, namely HepG2, MDA-MB-231, and BRIN-BD11 cells, in addition to livers of mice treated with statins for 12 weeks, leading us to conclude that neither short nor long term statin treatment alters cellular HDAC activity. We therefore considered the possibility that in live cells the statins may be unable to enter the nucleus, or that some other element of cellular activity may prevent them from interacting with HDAC proteins, as the studies by other groups were conducted either using recombinant HDAC proteins [13] or nuclear extracts [11,12]. To address this possibility, we conducted experiments using nuclear extracts, and yet again failed to find any HDAC inhibition by statins. The negative outcome of these experiments is unlikely to be due to inappropriately low statin concentration, as concentrations of up to 200 µM statins were used in the nuclear extract experiments. This concentration is significantly higher than the 10 µM doses reported to inhibit HDAC activity by Lin et al. [12] and well above the 1–15 nM mean serum concentration of statin users [25]. Furthermore, measurement of cellular cholesterol demonstrated that the statins did lower cellular cholesterol, indicating that the statins successfully entered the cells and inhibited HMG-CoA reductase. The contrast in results could also be due to the different cell types or the use of different HDAC substrates; previous studies used A549 lung carcinoma cells and primary glomerular mesangial cells, and all reported studies used the Fluor-De-Lys^®^ HDAC Activity Assay Kit. Concerning the latter, we also utilized the Fluor-De-Lys^®^ assay kit and found no effect with statins on HDAC activity in HepG2 or HeLa nuclear extracts. Concerning the former, we did not use the exact cell types previously used in the literature; however, we used a wider range of cell types than previous studies, and also examined liver tissue extracts; all cell lines responded as expected to established HDAC inhibitors. 

It could be argued that focusing on other specific cell or cancer types may have been a better approach, but in those circumstances, failure to find an effect would not preclude an effect in other cell types. Our initial studies were conducted in one cell type, namely HepG2 cells; once those studies were found to be negative, we took the approach of examining for effects in a range of cell types. We considered this to be appropriate on the basis of the mechanisms involved for agents with well-established effects such as butyrate and curcumin which mediate their effects by direct interaction with epigenetic enzymes and which are thus efficacious in most cell types. We reason that by using a variety of different cell types, we have shown there is a general lack of effect of statins on the enzymes examined. Furthermore, by conducting experiments using nuclear extracts we can deduce that statins do not inhibit activity through direct interaction with the HDAC enzyme, as posited by Lin et al. [12], while experiments with live cells and animal tissue suggest that statins do not alter HDAC activity indirectly through phenotypic or metabolic changes, although we concede that this cannot be ruled out in all circumstances.

In contrast to studies on HDACs, the effect of statins on HAT activity has not been previously reported. However, as with HDAC activity, our results were unable to demonstrate a direct effect of atorvastatin on HAT activity in HepG2 nuclear extracts.

There are some limitations to the use of fluorogenic HDAC substrates in activity assays. In particular, the Fluor-de-Lys™ substrate has been shown to interfere with HDAC activity and tends to overestimate inhibition of HDAC1 and HDAC6, while failing to detect inhibition of HDAC8 [26]. Similar experiments have not been reported for the cell permeable substrate used in the Sigma-Aldrich assay kit which we also used in our studies. However, the lack of HDAC and HAT inhibition in these assays is supported by the immunoassay results showing no increase in acetylation of histone H3K9 or multiple residues of histone H4. In fact, while Chen et al. [13] reported inhibition of recombinant HDAC1, HDAC2, and HDAC6 proteins by lovastatin and atorvastatin, they did not find an increase in histone H3 acetylation following 24 h treatment of A549 cells with 30–50 µM lovastatin. Conversely, Lin et al. [12] did report increased H3 acetylation following 16 h of treatment in A549 cells with 30 µM statins (lovastatin, simvastatin, pravastatin, fluvastatin, and atorvastatin).

Our results also showed that DNMT activity was not decreased by 24 h prior treatment with atorvastatin. By contrast, the opposite was found with an increase in DNMT activity. There is a caveat, since there a lack of an appropriate control inhibitor for this assay for studies on live cells and in the context of a control for a prolonged 24 h exposure, as explained later. However, our finding of no decreases in DNMT activity by statins in live cells is supported by a similar finding in cell extract experiments, where statins were added directly to cell extracts, and where we had a well-established DNMT inhibitor control, namely curcumin [27]. This enables us to conclude that statins do not directly inhibit DNMT activity. That 24 h curcumin treatment of cells did not reduce DNMT activity in live cells, has been observed elsewhere and is possibly due to degradation of curcumin in cell media [18]. Indeed the most well-established DNMT inhibitor—5-azacytidine (as well as other established DNMT inhibitors decitabine and zebularine)—does not interact with free DNMTs, instead inhibiting DNMT activity through its incorporation into DNA, and hence does not show inhibition in DNMT activity assays [28]. For this reason, combined with the marginal nonsignificance (*p* = 0.057) of the rationally designed inhibitor RG108 on cell extracts, we consider that there is a lack of appropriate established DNMT inhibitors to use in cell culture, which remains a challenge to be addressed in all such studies.

Our DNMT findings are in contrast with the results of Kodach et al. [14], who reported reduced DNMT activity in nuclear extracts from lovastatin treated HCT116 colon cancer cells. Several differences in design of our study and the latter study may account for the disparate results. Firstly, they used lovastatin, whereas we chose to use atorvastatin as our exemplar statin given the latter is the most commonly prescribed statin in numerous advanced economies including the United States [29], England [30], and Australia [31], while lovastatin is not available for therapeutic use in some regions. Additionally, Kodach et al. [14] used a different cell type and a longer treatment time, 48 h versus 24 h. Since these authors did not investigate if lovastatin directly inhibited DNMT enzyme activity, it cannot be ruled out that the observed decrease in activity is a result of changed phenotype, as they did observe increased differentiation and reduced ‘stemness’ of the cancer cells. Finally, different commercial kits were used, although both relied on similar principles based on use of an unmethylated DNA substrate and Adomet (S-adenosyl methionine) as a methyl donor, followed by measurement of methylation of the DNA of using a methylation specific antibody reporter system. 

## 4. Materials and Methods

### 4.1. Preparation of Stock Solutions of Treatments

Atorvastatin calcium salt, rosuvastatin calcium salt, simvastatin sodium salt and pravastatin sodium salt (Cayman Chemical, Ann Arbor, MI, USA), curcumin, and RG108 (Abcam, Cambridge, UK) were dissolved in dimethyl sulfoxide (DMSO) (Sigma-Aldrich, St. Louis, MO, USA) to a concentration of 10 mM and stored at −20 °C for short-term use or at −80 °C for long-term use. Sodium butyrate (Selleck Chemicals, Houston, TX, USA) was dissolved in ultrapure distilled water to a concentration of 50 mM and stored at −20 °C.

### 4.2. Cell Culture and Treatments

HepG2 human hepatocellular carcinoma cells were obtained from ATCC and provided by Dr Ross Graham (Curtin University, Perth, Australia). MDA-MB-231 human breast carcinoma cells were obtained from ATCC and provided by Professor Arunasalam Dharmarajan (Curtin University, Bentley, WA, Australia). BRIN-BD11 rat insulinoma cells were from Cell Bank Australia, deposited by Prof PR Flatt, University of Ulster, via the European Tissue Culture Collection. THP-1 human leukemic monocytes were obtained from Dr Hilary Warren (Canberra Hospital, Canberra, Australia) and provided via Professor Deirdre Coombe’s team (Curtin University, Bentley, WA, Australia). HepG2 cells were maintained in Dulbecco’s Modified Eagle’s medium: Ham’s F-12 Nutrient Mixture (DMEM: F12 media) (Sigma-Aldrich) supplemented with 10% fetal bovine serum (FBS) (Serana, Bunbury, WA, Australia). MDA-MB-231 cells, BRIN-BD11 cells, and THP-1 cells were maintained in Roswell Park Memorial Institute (RPMI) media (Sigma-Aldrich) supplemented with 10% FBS. All cells were maintained in 25-cm^2^ or 75-cm^2^ tissue culture flasks at 37 °C in a humidified incubator equilibrated with 5% CO_2_. All cells tested negative for mycoplasma contamination. Cells were seeded in 96-well plates or 25-cm^2^ or 75-cm^2^ tissue culture flasks and allowed to recover prior to treatment. Prior to use, THP-1 cells were differentiated into macrophages by the addition of 50 nM phorbol 12-myristate 13-acetate (PMA) (Sigma-Aldrich) to the culture media. THP-1 cells were differentiated for 48 h, then allowed to recover for 1 day before treatment. All cells were treated in media supplemented with 10% lipoprotein-deficient serum (LPDS) unless otherwise stated. Control cells were treated with DMSO as a vehicle control unless otherwise stated.

### 4.3. Preparation of Lipoprotein-Deficient Serum and High-Lipoprotein Serum

The density of FBS was adjusted to 1.21 g/mL with sodium bromide. The solution was transferred to 11.5 mL polyallomer ultracrimp tubes and centrifuged for 20 h at 70,000× *g* in a Sorvall WX ultracentrifuge. The base of the tube was pierced with a butterfly needle and 1 mL aliquots collected. The cholesterol and total protein of the aliquots was measured using the Amplex™ Red Cholesterol Assay Kit (Life Technologies, Carlsbad, CA, USA) and the Pierce BCA Protein Assay Kit (ThermoFisher, Waltham, MA, USA), respectively, to determine which aliquots were deficient in lipoproteins (LPDS) and those enriched in lipoproteins (HLPS). Salt was removed from the serum by centrifugation with Zeba™ desalting spin columns 7 K MWCO (ThermoFisher) and serum was sterilized with a Millex 0.22 um syringe filter prior to storage at −20 °C.

### 4.4. Animal Studies

Eight-week-old male C57B1/6J mice from the Animal Resource Centre, Murdoch, Western Australia were delivered to the animal facility, Curtin University. Following acclimatization, mice were randomly assigned to either normal diet (ND) or high-fat diet (HFD) (Specialty Feeds, Glen Forrest, WA, Australia). From week 4, mice were further divided into treatment groups to receive either 10 mg/kg/day of atorvastatin or vehicle (water) by gastric gavage for a further 12 weeks. Following completion of the treatment period, mice were starved for 6 h and anesthetized in an isoflurane chamber before being euthanized by cervical dislocation. Liver samples were removed from the carcass and collected in prechilled microcentrifuge tubes and snap-frozen either in dry ice or liquid nitrogen. Animal experiments were approved by Curtin University’s Animal Ethics Committee (AEC_2016_17, approval date 14 April 2016).

### 4.5. Nuclear Extraction

Nuclear extraction was conducted using the Abcam Nuclear Extraction Kit according to the manufacturer’s instructions. For experiments on cultured cells, confluent cells were scraped and incubated on ice for 10 min in pre-extraction buffer containing dithiothreitol (DTT) and protease inhibitor cocktail (PIC), then centrifuged at 12,000 rpm for 1 min. The cytoplasmic supernatant was removed, and the pellets incubated on ice for 15 min in nuclear extraction buffer containing DTT and PIC, then sonicated 3 times for 10 s. The extract was then centrifuged at 14,000 rpm for 10 min at 4 °C and the supernatant removed for protein quantification using the Pierce Coomassie Protein Assay Kit (ThermoFisher). For experiments on murine liver, samples were placed in a glass homogenizer containing pre-extraction buffer with DTT and PIC and homogenized manually. Homogenized samples were transferred to microcentrifuge tubes and incubated on ice for 15 min then centrifuged at 10,000 rpm for 10 min at 4 °C. The cytoplasmic supernatant was removed, and nuclear extraction continued as for cultured cells.

### 4.6. Whole Cell Protein Extraction

Whole cell extractions were only conducted on cultured cells. Confluent cells were scraped then lysed on ice for 20 min in radioimmunoprecipitation assay (RIPA) lysis buffer (Sigma-Aldrich) containing protease and phosphatase inhibitors (Sigma-Aldrich). The lysate was sonicated 10 times for 10 s then centrifuged at 14,000 rpm for 10 min and the protein rich supernatant was removed for protein quantification using the Pierce BCA Protein Assay Kit (ThermoFisher).

### 4.7. Viability Assays

Viability was determined using alamarBlue^®^ (ThermoFisher) as a measure of total cellular metabolism. Following 22 h treatment, 10 µL of alamarBlue^®^ was added to the treatment media and cells were incubated for a further two hours. Fluorescence was then read by the EnSpire Multimode Plate Reader (PerkinElmer, Waltham, MA, USA) with an excitation wavelength of 540 nm and an emission wavelength of 590 nm.

### 4.8. Cellular Cholesterol Content

After 24 h of statin treatment, the culture medium was removed and lipids were extracted using 3:2 hexane:isopropanol. Lipids were transferred to a 96 V well plate and allowed to dry at room temperature. Cholesterol content was assessed using the Amplex Red Cholesterol Assay Kit (Life Technologies). Samples were dissolved in a reaction buffer and isopropanol mixture (1:1) and incubated with catalase for 15 min at 37 °C to eliminate endogenous peroxides which would otherwise interfere in this assay [32]. Amplex Red working solution was added and following 30 min incubation, fluorescence was read with an excitation wavelength of 540 nm and an emission wavelength of 590 nm.

### 4.9. HDAC Activity

The In Situ Histone Deacetylase (HDAC) Activity Fluorometric Assay Kit (Sigma-Aldrich) was used to measure HDAC activity according to the manufacturer’s instructions. The HDAC substrate was added to wells containing live cells directly to the treatment media after 23 h treatment. For HDAC activity assays on the nuclear extracts, enzyme substrate was added concurrently with statin treatments in phosphate-buffered saline. After 1-h incubation, the HDAC developer was added and the plate was incubated for a further 30 min and fluorescence read with an excitation wavelength of 368 nm and an emission wavelength of 442 nm.

Confirmatory experiments were carried out with the Fluor De Lys^®^ HDAC fluorometric activity assay kit (Enzo Life Sciences, Farmingdale, NY, USA) according to the manufacturer’s instructions. HeLa nuclear extract provided with the kit and HepG2 nuclear extract prepared as above were incubated with treatments and substrate for 30 min. Following incubation with developer for 10 min, fluorescence was read with an excitation wavelength of 360 nm and an emission wavelength of 460 nm.

### 4.10. HAT Activity Assays

HAT activity was measured with the EpiQuik™ HAT Activity/Inhibition Assay Kit (Epigentek, Farmingdale, NY, USA) according to the manufacturer’s instructions. The HAT substrate was captured in strip wells for 45 min. Following washing, strip wells were incubated with acetyl-CoA, HepG2 nuclear extracts, and treatments for 90 min at 37 °C. Following washing, wells were incubated sequentially with the capture antibody and detection antibody, followed by the addition of the developer solution. Once the solution in the control wells changed to a medium blue, the enzymatic reaction was stopped with the stop solution and the absorbance read at 450 nm.

### 4.11. DNMT Activity Assays

DNMT activity was measured with the Abcam DNMT Activity Assay Kit according to the manufacturer’s instructions, with the following modification; whole cell extracts were used as opposed to nuclear extracts since experiments with nuclear extracts were not successful. For direct DNMT inhibition experiments cell extracts were incubated with 200 µM atorvastatin, curcumin or RG-108 and the Adomet methyl donor at 37 °C for 2 h in strip wells coated with DNMT substrate. For treated cells, equal amounts of total protein were incubated with Adomet. Following incubation and washing, wells were incubated sequentially with the capture antibody, detection antibody, and enhancer solution, followed by the addition of the developer solution. Once the solution in the control wells changed to a medium blue, the enzymatic reaction was stopped with the stop solution and the absorbance read at 450 nm with a reference wavelength of 655 nm.

### 4.12. Immunoblotting for Modified Histones

Equal amounts of protein were fractionated on Bolt™ 4–12% Bis-Tris Plus Gels (ThermoFisher) following denaturing at 98 °C for 10 min. Proteins were transferred onto nitrocellulose membranes using the iBlot Gel Transfer Device (Invitrogen, Carlsbad, CA, USA) and blocked with 3% bovine serum albumin (Bovogen, Keilor East, VIC, Australia) in Tris-buffered saline with 0.1% Tween 20 (TBST) for at least 1 h. Membranes were incubated overnight with primary antibodies against acetylated H3K9 and acetylated H4 (S1, K5, K8, and K12) (Santa Cruz, Dallas, TX, USA) diluted in blocking buffer. GAPDH (Abcam) was used as a housekeeping reference protein. Membranes were incubated with goat anti-mouse IgG H&L horseradish peroxidase (HRP) (Abcam) in blocking buffer for at least one hour, then washed 3 times with TBST. Immunodetection was performed with the Amersham ECL Prime Western Blotting Detection Reagent (GE Healthcare, Chicago, IL, USA) in the Chemi-Doc™ Gel Imaging System (Bio-Rad, Hercules, CA, USA). Band density was measured with Image Lab 6.0.1 software (Bio-Rad) and normalized to the density of housekeeping protein GAPDH or total protein as determined by Coomassie Blue staining.

### 4.13. Statistical Analysis

At least three independent experiments were conducted. Statistical significance was determined with analysis of variance (ANOVA) with results considered significant if *p* < 0.05 using GraphPad Prism 8.0.1 software (GraphPad Software, San Diego, CA, USA).

## 5. Conclusions

There is substantial interest in inhibitors of epigenetic enzymes as anticancer agents: that commonly prescribed drugs, such as statins, may confer anticancer activities via epigenetic mechanisms is therefore an attractive proposition. However, our results indicate that statins are unlikely to inhibit the three major classes of epigenetic modifying enzymes, namely HDACs, HATs, and DNMTs, and in fact may increase DNMT activity. Our comprehensive studies of HDAC activity in multiple cell types, in both live cells and nuclear extracts and using two different HDAC substrates demonstrate no effect with statin treatment. While our in vitro studies indicate it is unlikely that statins directly inhibit these epigenetic enzymes, it is feasible that through effects on cell metabolism and/or phenotype, statins may have an indirect effect as described previously for metformin, particularly with prolonged treatment. However, in the context of HDACs, our studies on the livers of statin treated mice would also seem to rule out effects of prolonged treatment on HDAC activity, direct or otherwise. It cannot be ruled out that statins may influence subclasses of these various enzymes or the recruitment of enzymes to certain genomic regions without altering the overall activity. Thus, statins may influence epigenetic modifications in the proximity of certain genes and thereby affect gene expression through epigenetic means. To that end, targeted studies of specific genomic regions are warranted and are currently under investigation.

## Figures and Tables

**Figure 1 cancers-11-00516-f001:**
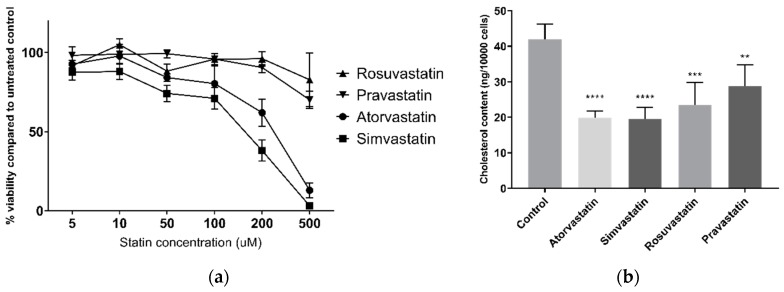
(**a**) HepG2 cell viability following 24 h statin treatment; (**b**) HepG2 cellular cholesterol following 24 h statin treatment. ** *p* < 0.01, *** *p* < 0.001, **** *p* < 0.0001 compared to control.

**Figure 2 cancers-11-00516-f002:**
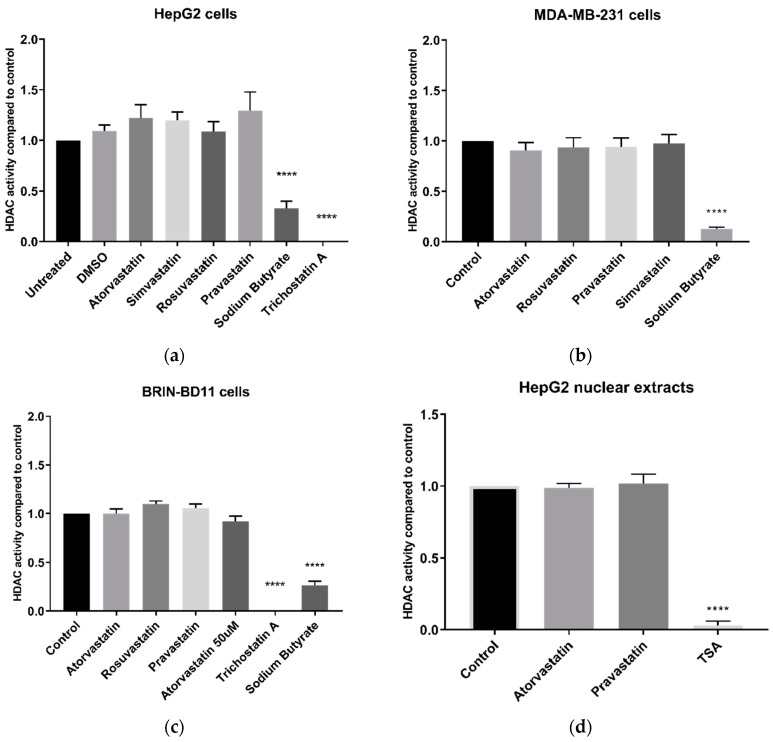
HDAC activity using the Sigma-Aldrich In Situ HDAC activity kit. (**a**) HDAC activity of live HepG2 cells following 24 h treatment with 10 µM statins; (**b**) HDAC activity of live MDA-MB-231 cells following 24 h treatment with 10 µM statins; (**c**) HDAC activity of live BRIN-BD11 cells following 24 h treatment with 10 µM statins; and (**d**) HDAC activity following direct application of 100 µM statins to HepG2 nuclear extracts. **** *p* < 0.0001 compared to control.

**Figure 3 cancers-11-00516-f003:**
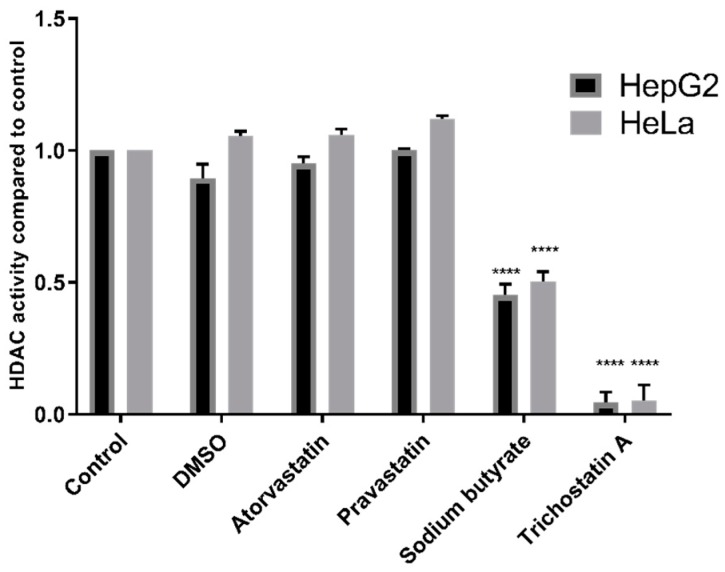
HDAC activity of HepG2 and HeLa nuclear extracts treated with 200 µM statins, 1 mM sodium butyrate, or 4 µM TSA using the Fluor De Lys^®^ HDAC activity kit. **** *p* < 0.0001 compared to control.

**Figure 4 cancers-11-00516-f004:**
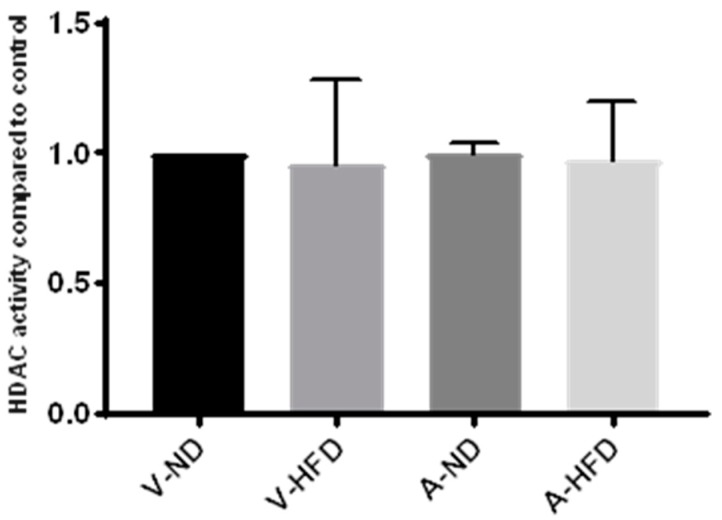
HDAC activity in nuclear extracts from livers of mice on a normal diet (ND) or high-fat diet (HFD) treated with atorvastatin (A) or water (V) for 12 weeks. *n* = 3 per group.

**Figure 5 cancers-11-00516-f005:**
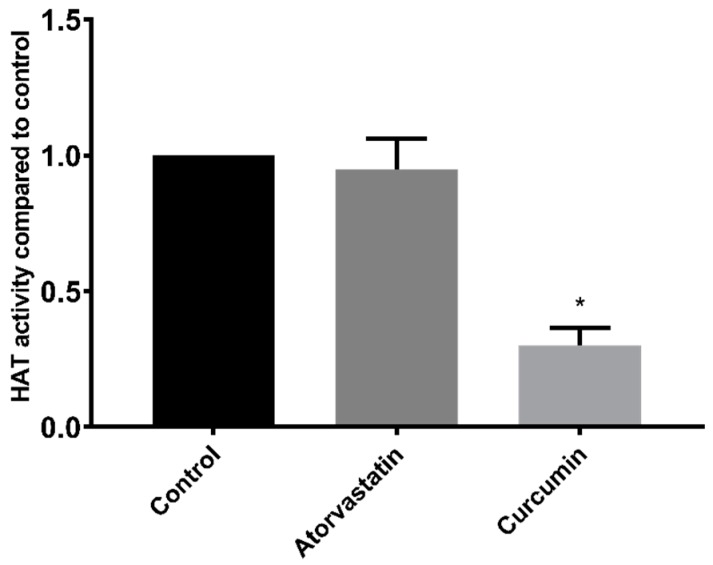
HAT activity of HepG2 nuclear extracts treated with 100 µM atorvastatin or curcumin. * *p* < 0.05 compared to control.

**Figure 6 cancers-11-00516-f006:**
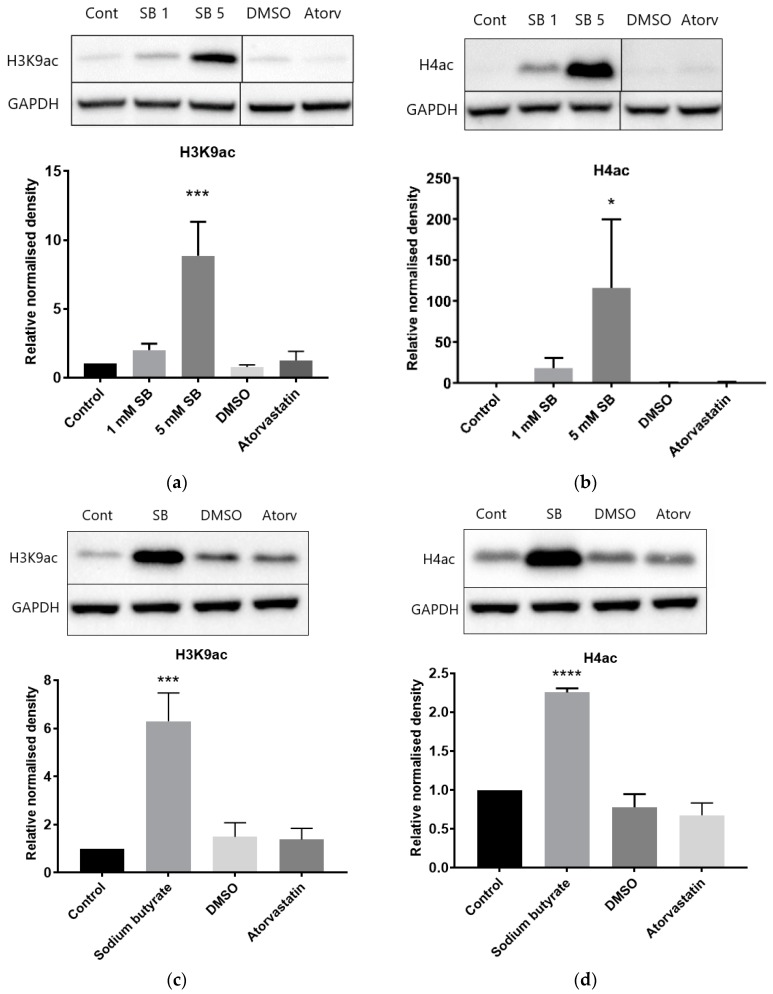
Influence of atorvastatin on histone acetylation in HepG2 and THP-1 cell extracts by Western blot analysis. (**a**) Relative level of acetylated histone H3K9 in HepG2 cells; (**b**) relative level of acetylated histone H4 in HepG2 cells; (**c**) relative level of acetylated histone H3K9 in THP-1 cells; and (**d**) relative level of acetylated histone H4 in THP-1 cells. Cells were treated for 24 h with 1 mM or 5 mM sodium butyrate (SB), 10 µM atorvastatin, or equivalent DMSO vehicle control. The graphs illustrate combined density readings normalized to GAPDH from three independent experiments. * *p* < 0.05, *** *p* < 0.001, **** *p* < 0.0001.

**Figure 7 cancers-11-00516-f007:**
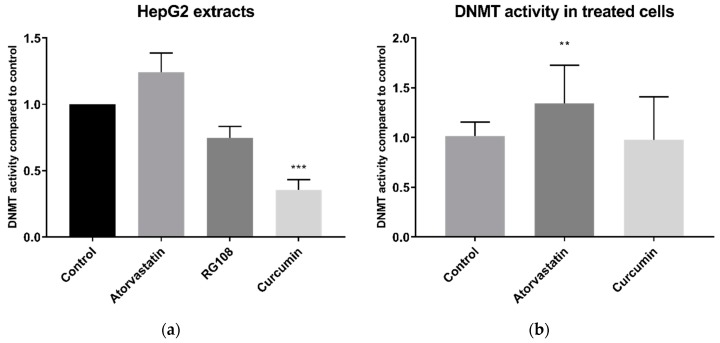
DNMT activity in HepG2 whole cell extracts. (a) DNMT activity in whole cell extracts treated directly with 200 µM atorvastatin, RG-108 or curcumin. (b) DNMT activity in whole cell extracts from cells treated for 24 h with 10 µM atorvastatin or curcumin. ** *p* < 0.01, *** *p* < 0.001

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
