# Peer review of "Statins Do Not Directly Inhibit the Activity of Major Epigenetic Modifying Enzymes"

_cancers, 2019, doi:10.3390/cancers11040516_

Reviewer 1 Report

The article by Bridgeman and colleagues, "Statins do not directly inhibit the activity of major epigenetic modifying enzymes", clearly presents results assessing how commonly used statin drugs affect HDACs and HATs. They conclude that statins affect neither HDAC no HAT activity and use proper positive and negative controls for the assays. They also test whether DNMT activity is affected; unfortunately these results are much less clear because a positive control (for example, DNMT inhibitor) is not used. This control is necessary to make a conclusion about the effects of statins on DNMTs. 

Author Response

The report by Reviewer 1 is largely positive, with the only concern being the lack of a positive control for DNMT inhibition.

 Nevertheless, we would like to comment on the DNMT studies: for clarification, we did have a positive control, namely curcumin; and for our studies on cell extracts, this was shown to inhibit DNMT activity, as expected. No effect was seen for curcumin when added to cultured cells, but it is known that under such conditions (24 hours in culture medium), curcumin was less effective, due to instability in culture media as referenced in the article. We could have excluded that data, but included it for transparency. Others have also observed such a phenomena with curcumin-that is the lack of an effect in cultured cells (Link et al, 2013; 8(2):e57709, PLoS One). The question then arises as to whether the work on the cell extracts, where curcumin did have an effect is relevant. In our opinion we consider that it is, since the mechanism by which curcumin operates is by direct inhibition of the enzyme (e.g by binding to it's catalytic site) and no via indirect mechanisms which would be more relevant to work on cultured cells.

Furthermore it should be noted that the most well-established DNMT inhibitor, 5-azacytidine (as well as other established DNMT inhibitors decitabine and zebularine), does not interact with free DNMTs, instead inhibiting DNMT activity through its incorporation into DNA, and hence does not show inhibition in DNMT activity assays (Brueckner et al, 2005; 65, Cancer Research). For this reason, combined with the non-significance of the rationally designed inhibitor RG-108 on cell extracts, we feel their is a lack of appropriate established DNMT inhibitors suitable for use in cell culture studies of this type.

 Finally, the aspect on DNMT is but one of the many aspects examined, and we wish to emphasize that the study conducted is the most comprehensive one to date on statins.

 Reviewer 2 Report

Although the manuscript is interesting and certainly complete with well-conducted experiments, the conclusions of the authors are very approximate. Since other experimental evidences contrast with their results, maybe they should focus on studying a specific cancer model. I suggest investigating why statins have no effect on epigenetic modulators in hepatocellular carcinoma performing the experiments on different cell lines, not only HepG2 cells. In the article “Altered microRNome Profiling in Statin-Induced HepG2 Cells: A Pilot Study Identifying Potential new Biomarkers Involved in Lipid-Lowering Treatment”, for example, the authors concluded that in HepG2 cells two statins modulated many miRs but the biological consequences of this are to be determined. The triple negative breast cancer is an other model very interesting to study but even in this case only one cell line is not sufficient.

Author Response

We thank the reviewer for their comments ("the manuscript is interesting and certainly complete with well-conducted experiments"), and the rating of "yes" for every category (introduction, research design, methods description, results presentation, and conclusion).

 We do note some concerns: "that the conclusions of the authors are very approximate. Since other experimental evidences contrast with their results, maybe they should focus on studying a specific cancer model".

Concerning the latter, we consider this to be a philosophical point on how best to approach such investigations: each approach, ours or that suggested by the reviewer have their advantages and disadvantages.

 For example, if focusing on one  specific cell or cancer type, the failure to find an effect would not preclude an effect in other cell types. Once our initial studies in one cell type (HepG2) were found to be negative, we therefore took the philosophical view of examining for effects in a range of cell types. We considered this to be appropriate on the basis of the mechanisms involved for agents with well established effects such as butyrate and curcumin, which  mediate their effects by direct interaction with epeginetic enzymes, for example, and respectivey, HDACs and DNMT.

 Of note, this is the most comprehensive study to date on the epigenetic effects of statins and contrast with all previous studies which were only conducted on the one cell type or extract of. For example, in the case of HDAC activity, we demonstrated no effect in 4 cell lines plus an animal study. Of note, the mechanisms invoked in those studies, that is those studies which are contradicted by ours,  was one of direct HDAC inhibition by statins, the very question we examined, and using similar techniques.

We respectfully suggest that there would be interest in our more comprehensive studies.

Round  2

Reviewer 1 Report

The authors have addressed my concerns and I believe the manuscript is suitable for publication.

Reviewer 2 Report

Not any, I recommend the manuscript for publication in Cancers.